# Mining cancer genomes for change-of-metabolic-function mutations

Kevin J. Tu [1,2,3], Bill H. Diplas[4], Joshua A. Regal[1], Matthew S. Waitkus [5], Christopher J. Pirozzi[6] & Zachary J. Reitman [1,5,6 ✉]

Enzymes with novel functions are needed to enable new organic synthesis techniques. Drawing inspiration from gain-of-function cancer mutations that functionally alter proteins and affect cellular metabolism, we developed METIS (Mutated Enzymes from Tumors In silico Screen). METIS identifies metabolism-altering cancer mutations using mutation recurrence rates and protein structure. We used METIS to screen 298,517 cancer mutations and identify 48 candidate mutations, including those previously identified to alter enzymatic function. Unbiased metabolomic profiling of cells exogenously expressing a candidate mutant (OGDHLp.A400T) supports an altered phenotype that boosts in vitro production of xanthosine, a pharmacologically useful chemical that is currently produced using unsustainable, water-intensive methods. We then applied METIS to 49 million cancer mutations, yielding a refined set of candidates that may impart novel enzymatic functions or contribute to tumor progression. Thus, METIS can be used to identify and catalog potentially-useful cancer mutations for green chemistry and therapeutic applications.

[1] Department of Radiation Oncology, Duke University, Durham, NC 27710, USA. [2] Department of Cell Biology and Molecular Genetics, University of Maryland, College Park, MD 21044, USA. [3] Cancer Research UK Cambridge Institute, Li Ka Shing Centre, University of Cambridge, Cambridge CB2 0RE, UK. [4] Department of Radiation Oncology, Memorial Sloan Kettering Cancer Center, New York, NY 10065, USA. [5] Department of Neurosurgery, Duke University, Durham, NC 27710, USA. [6] Department of Pathology, Duke University, Durham, NC 27710, USA. ✉email: zjr@duke.edu

Cancer arises through a process of clonal evolution that selects for genetic alterations of net benefit to tumor cells. This evolutionary process occasionally selects for "change of metabolic function" (COMF) mutations that affect enzymes and other proteins that regulate cellular metabolism. A notable example of COMF is that of the p.R132H mutation in cytosolic NADP-dependent isocitrate dehydrogenase (IDH1), identified in up to 90% of certain brain tumor subtypes and other cancer types[1,2]. While originally hypothesized to be a loss-of-function variant[3], unbiased metabolite profiling experiments ultimately revealed that IDH1p.R132H exhibited COMF activity that catalyzes the conversion of α-ketoglutarate to R-2-hydroxyglutarate[4]. The accumulation of R-2-hydroxyglutarate causes widespread epigenetic reprogramming that likely drives oncogenesis[5–7].

The IDH1 case highlights important insights to identify cancer-derived COMF mutations, including: (i) focusing on mutations that are recurrent, (ii) concentrating on mutations in functional enzyme structures such as the active site (iii) using unbiased metabolite profiling to reveal unexpected gained functions.

Production of organic chemicals often rely on synthetic approaches; however, toxic chemicals and petroleum-based sources are often required and many natural products are too complex for conventional synthesis. Therefore, enzymes with new catalytic activities are needed to enable new organic synthesis methods and metabolic engineering techniques[8,9]. We previously showed that cancer-associated COMF mutations can guide enzyme redesign to generate novel, immediately useful catalytic activities. We previously applied the IDH1 COMF mutations to the distantly related yeast and bacteria homoisocitrate dehydrogenase enzymes[10]. IDH1 and homoisocitrate dehydrogenase differ in that IDH1 accepts the 5-carbon α-ketoglutarate substrate while the homoisocitrate dehydrogenases accept the 6-carbon α-ketoadipate substrate. Applying the R132H mutation from IDH1 to homoisocitrate dehydrogenase conferred a novel catalytic function to convert α-ketoadipate to (R)-2-hydroxyadipate. Such a catalytic function had not previously been found in nature. Further, this catalytic function was sought after to enable a bio-based method for the production of adipic acid, a valuable commodity chemical used to synthesize nylon that erstwhile required fossil fuel substrates for synthesis[11]. Thus, we showed that cancer-derived mutations could be applied to generate useful new catalytic activities and industrial chemistry applications.

Cancer mutational data, therefore, represents a source of functional diversity to identify mutant proteins with novel functions, as well as motifs that may be amenable to synthetic enzyme redesign. These mutations may be used to create new biocatalysts for a variety of fields. Cancer mutational data has accumulated at exponential rates with improvements in sequencing technology[12]. Indeed, more than 11,000 cancer tissues have now been analyzed by genome-wide sequencing approaches via efforts such as The Cancer Genome Atlas in the decade since our initial report of cancer-guided enzyme redesign based on IDH1 mutations[10]. However, prediction of mutations that lead to COMF through current variant prioritization strategies and strictly computational predictions is challenging[13]. Here, we sought to mine cancer mutational data for yet-undiscovered COMF mutations that may confer useful new functions to enzymes. For instance, we were particularly interested in catalytic functions that could enable new organic synthesis routes, provide useful steps in metabolic engineering processes, and/or provide new tools for fine chiral chemical production[14].

In this report, we develop a bioinformatic pipeline to screen genome-wide cancer sequencing data for candidate COMF mutations (graphical representation in Supplementary Fig. 1). We named this approach METIS (Mutated Enzymes from Tumor In

silico Screens). We applied METIS to a comprehensive catalog of cancer mutations. We then generated a metabolomic profiling dataset to identify metabolic perturbations that may be caused by four top candidates from this screen. This dataset identified cellular metabolites uniquely perturbed by the candidate COMF mutants but not by their wild-type controls, suggesting that the mutants confer a COMF that affected the cellular metabolome. The current work presents a pipeline to identify candidate COMF mutants from cancer sequencing data, provides a catalog of nominated candidates, and shows a use-case for unbiased metabolite profiling to identify mutation functions.

## Results

**METIS1: a pipeline to identify COMF cancer mutations.** Our approach to identify candidate enzyme COMF mutations from somatic cancer mutational data is outlined in Fig. 1 and described in the Methods. METIS primarily relies on the ability to screen for recurrent mutations that occur separately in multiple unique patients' tumors, indicating that they are likely to be functional.

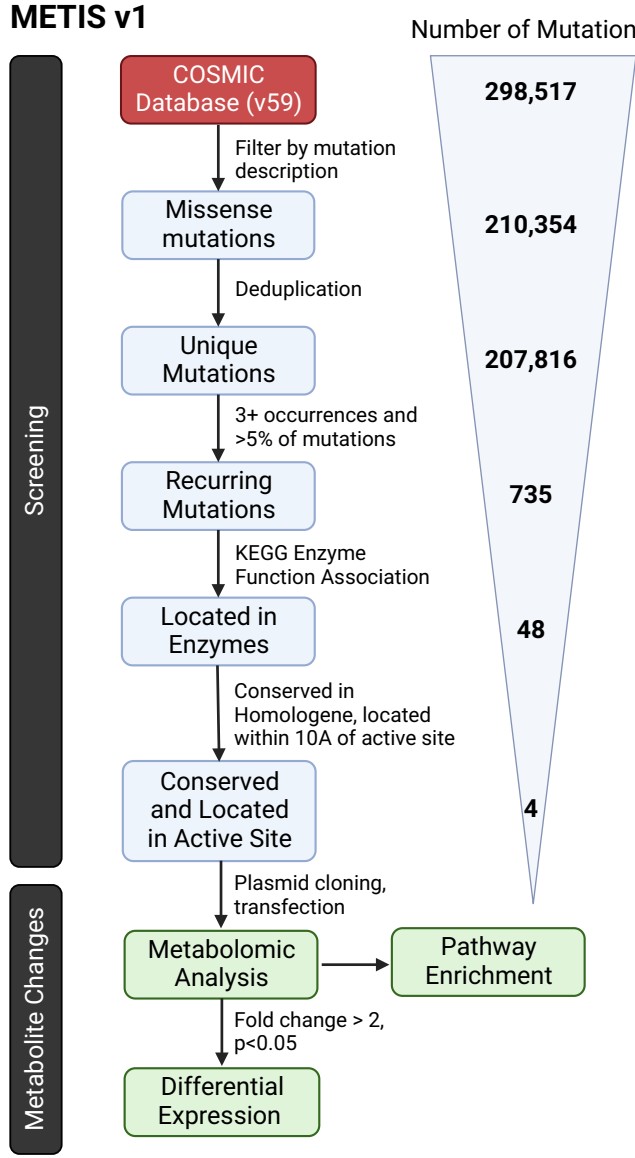

**Fig. 1 METIS pipeline to identify COMF mutations in cancer mutational data.** Schematic showing pipeline, with number of mutations filtered at each step.

We created a mutation filtering scheme based on the assumptions that COMF mutations are:

1. Recurrent[15]
2. Missense mutation[16]
3. Have an "oncogene" mutation distribution[17]
4. Located in highly-conserved residues[18]
5. Located in enzyme active site[19]

We sought to specifically identify mutational distributions that were likely to confer a COMF by occurring in functional enzyme active site structures or similarly-important structures. We initially applied METIS to the Catalog of Somatic Mutations in Cancer (COSMIC) dataset of mutations identified in genome-wide and panel-based cancer sequencing studies (COSMIC v59). Out of the 210,354 missense mutations in the dataset, we filtered in mutations that occurred in ≥3 patients and also accounted for >5% of all mutations in a given gene (Supplementary Note). We further filtered out duplicate mutation entries, mutations that were likely to be germline polymorphisms, and multiple tumor samples from the same patient (see Methods). After this analysis, 735 mutations were identified in 125 genes (details of filtered-out mutation sites in Supplementary Table 1).

We next sought to identify recurrent enzyme mutations most likely to confer novel COMF activities based on known structural, functional, and evolutionary information. We focused on the most frequent recurrent mutation in each gene. Mutations in proteins with associated enzymatic function were identified based on an associated Enzyme Commission number for the gene product. The 48 recurrent enzyme mutations and considerations for including or excluding them from further analysis are cataloged in Supplementary Table 2. As expected, the majority of these enzyme mutations (31/48) were previously known to be frequently mutated in cancer and thought to confer gain-of-function activities (for instance, mutations in *IDH1p.R132H, IDH2p.R140Q, EZH2p.Y646F, PIK3CAp.H1047R,* and *BRAFp.V600E*)[20–23]. This finding confirms that our pipeline is able to identify gain-of-function enzyme mutations in cancer data. In particular, when compared to a gold-standard panel of previously identified COMFs, the sensitivity of METIS was 86% given its ability to detect most (36/42) known COMF mutations (Supplementary Tables 2–4). We also estimated the specificity at 95% given the high number of yet-unestablished COMF mutations and enzymes involved in metabolic pathways. Overall accuracy was estimated at 93% (282/300). This estimate suggests that METIS1 was reasonably effective at identifying known COMF mutations.

Since we sought to identify novel COMF mutations, the known gain-of-function mutations were filtered out leaving 17 enzyme-coding genes associated with recurrent cancer mutations. We next reasoned that novel COMF mutations would affect residues that are highly conserved based on the examination of multiple sequence alignment data in HomoloGene. Additional filtering steps were taken to focus on enzymes that would be feasible to manipulate in downstream biochemical analyses (ie, <1500 amino acid residues). We also removed candidates that appeared to result from the incorrect annotation of pseudogenes or germline polymorphisms. For the remaining potential candidates, we examined available 3D structural data in the Protein Data Bank (PDB) for the encoded proteins or available homologs. We reasoned that mutations likely to confer a COMF would reside in or near the enzyme active site where mutations alter ligand-binding functions, as in the case of IDH1 and other known COMF mutations. Therefore we retained only those mutations that affect residues within 10 Å of the active site or ligand binding sites. After these filtering steps, four candidate COMF mutations were obtained (Supplementary Table 3).

Structural features, cancer mutation analysis, and homology analysis of the top four candidates are detailed further in our Supplementary Note. We also conducted a rational literature review to carefully assess the biological functions of the COMF candidates in the context of cancer. The candidate COMF mutations included a p.Y371H mutation in the E3 Ubiquitin ligase Cas-Br-M ecotropic retroviral transforming sequence (CBL, Fig. 2a–c), a recurrent p.R228C mutation in the protein-UDP acetylgalactosyltransferase named Williams-Beuren syndrome chromosome region 17 (WBSCR17, now renamed to GALNT17, Fig. 2d–f), a recurrent p.R364C mutation in the anion/sugar transporter Solute Carrier Family 17 (SLC17A5), also known as Sialin (Fig. 2g, h), and recurrent p.A400T mutations in oxoglutarate dehydrogenase-like (OGDHL) which likely encodes an additional isoform of α-ketoglutarate dehydrogenase, which converts α-ketoglutarate to succinyl-CoA and produces NADH for the respiratory chain[24] (Fig. 2i–k). The CBLp.Y371H and OGDHLp.A400T mutations have been noted to contribute to cancer development through cytokine-independent growth and metabolic reprogramming[25,26].

**Unbiased global metabolite profiling of candidate COMF mutants**. We hypothesized that the expression of one or more COMF candidates would cause metabolic changes in cancer cells that could be detected by global metabolite profiling approaches. Our screening approach identified candidate COMF mutations primarily based on their genetic distribution in cancer and is otherwise agnostic to their function. Therefore, the function that may be conferred by any given candidate COMF mutation was not known a priori. To test if any of the candidate mutations may have a COMF and to provide insights on any putative COMF, we executed a metabolomic screen (schematic in Fig. 3a). We first expressed the top four candidate COMF mutants in HeLa cells (CBL p.Y371H, WBSCR17 p.R228C, SLC17A5 p.R364C, OGDHL p.A400T). As controls, we compared these to wild-type versions (CBL-WT, WBSCR17-WT, SLC17A5-WT, OGDHL-WT, respectively). Further, we included empty vector controls and a control in which a nonfunctional protein GFP was expressed.

The metabolite profiling data were first analyzed by identifying metabolites that were significantly altered in each group after stringent false discovery (FDR) correction. 277 metabolites were semi-quantified, including 231 with known composition and 46 unique biochemicals with unknown composition (Fig. 3b, Supplementary Data 1, 2). Each experimental group ($n = 3$ samples per group) was compared to all other groups and mean ion count fold change, p-values, and FDR-adjusted q-values were calculated (Supplementary Data 3).

Importantly, our negative EV and GFP controls were not associated with significantly altered metabolites. No biochemicals were considered significantly altered in the CBL mutant, WBSCR17 mutant, or SLC17A5 mutant groups after FDR correction.

OGDHL p.A400T was the only COMF candidate associated with significant changes after FDR correction (q < 0.05, two-tailed Welch's unequal variances t-test with Bonferroni correction). Xanthosine, a key intermediate in purine metabolism, was increased 2.9-fold in the OGDHL p.A400T group compared to all other groups ($P = 1.2 \times 10^{-9}$, $q = 3.2 \times 10^{-7}$, Fig. 3c). Flavin mononucleotide (FMN) was also increased 1.8-fold in the OGDHL p.A400T group compared to all others ($P = 1.2 \times 10^{-5}$, $q = 0.003$). Several biochemicals were significantly altered in the wild-type groups, including 26 biochemicals in the OGDHL WT group (q < 0.05 for each, see Metabolic Pathway Analysis section below). Thus, unbiased metabolite profiling revealed highly significant changes in xanthosine and

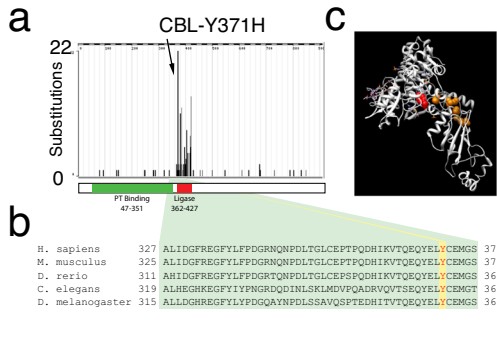

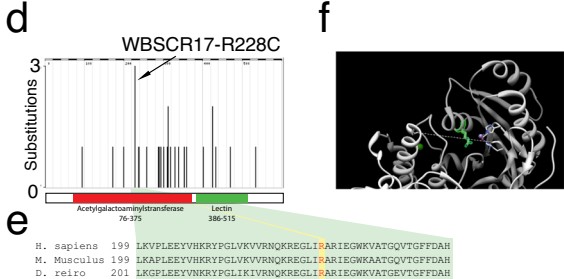

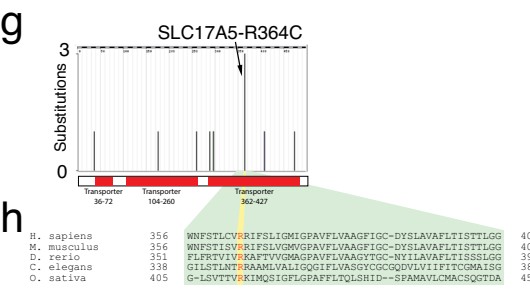

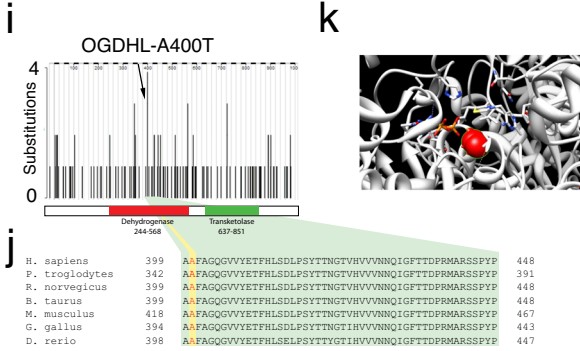

**Fig. 2 COMF candidates nominated by METIS. a** CBL mutational distribution in COSMIC v59 showing recurrent Y371H mutations. **b** Multiple alignment with Y371 shown in red. **c** Structure of human CBL-UBCH7 complex (PDB: 1FBV) with Y371 shown in red, other recurrently-mutated residues shown in orange, and ligands shown in stick representations. **d** WBSCR17 mutational distribution in COSMIC v59 showing recurrent R228C mutations. **e** Homology alignment for WBSCR17 with R228 shown in red. **f** Structure of mouse homolog of WBSCR17 (PDB: 1XHB) with R228 in active site pocket shown in green. **g** SLC17A5 mutational distribution in COSMIC v59 with recurrent R364C mutations. **h** Multiple alignment of SLC17A5 R364 region with R364 shown in red. **i** OGDHL mutational distribution in COSMIC v59 with recurrent A400T mutations shown. **j** Multiple alignment of OGDHL with A400 shown in red. **k** Structure of SucA domain of Mycobacterium smegmatis alpha-ketoglutarate decarboxylase in complex with acetyl-CoA (PDB: 2XTA) with residue homologous to A400 shown in red with ligand shown in stick representation. Primary enzymatic function domains are represented by red boxes while secondary functions are represented by green boxes underneath the mutation plot.

FMN associated with the expression of a candidate COMF mutation in OGDHL.

**Metabolic pathway analysis.** We next examined metabolic pathways in the global metabolite profiling data for each candidate COMF mutant and directly compared each COMF candidate to its respective WT control. We noted pathway-level alterations in amino acid metabolism associated with CBL p.Y371H (Fig. 4a and Supplementary Fig. 2), lipid metabolism associated with WBSCR17 p.R228C (Fig. 4b and Supplementary Fig. 3), and nucleoside metabolism associated with SCL17A5 p.R364C (Fig. 4c and Supplementary Fig. 4), which are detailed further in the Supplementary Note.

OGDHL WT caused significant increases in amino acid metabolism (15 biochemicals with q < 0.05 compared to all other groups) inducing increases in 11 of the 20 standard amino acids such as glutamine, glutamate, phenylalanine, tyrosine, tryptophan, leucine, valine, methionine, isoleucine, and proline (Supplementary Fig. 5a). OGHDL WT also caused significant increases in nucleoside metabolism including in cytidine triphosphate, uridine 5'-triphosphate, and uridine 5'-diphosphate (Supplementary Fig. 5b). Other metabolites significantly increased in the OGDHL WT group included phosphate, choline phosphate, and eight unknown compounds (Supplementary Fig. 5c). The changes in amino acid and nucleoside metabolism may relate to OGDHL's role in metabolizing α-ketoglutarate, which is reversibly converted by glutamate dehydrogenase to glutamate. Glutamate can then be directly or indirectly converted to other amino acids and nucleosides. Thus, OGDHL WT exerts

significant changes on biochemicals that may be related to its normal enzymatic functions[24].

OGDHL p.A400T elicited unique metabolic alterations compared to OGDHL WT, suggesting that this mutant confers a metabolic COMF. When comparing each group with all other groups, none of the amino acids or nucleosides altered in the OGDHL WT group were appreciably altered by OGDHL p.A400T (Fig. 3b). However, similar to the global analysis, OGDHL p.A400T is associated with a 2.6-fold increase in xanthosine compared to OGDHL WT (P = 0.02). Xanthosine is a key intermediate in the purine degradation pathway. Xanthosine can be phosphorylated to form xanthosine monophosphate (XMP). XMP is metabolized to guanosine monophosphate (GMP) by GMP synthetase, which involves the conversion of glutamine and ATP to glutamate and ADP and free pyrophosphate as part of the reaction. OGDHL p.A400T may affect glutamine, which was downregulated 0.23-fold (P = 0.058), by metabolizing α-ketoglutarate, leading to deregulation of GMP synthase function to lead to xanthosine accumulation (Fig. 3c). An increase in FMN was seen when comparing OGDHL p.A400T to all other samples, although this was not statistically significant when comparing OGDHL p.A400T only to OGDHL WT (Supplementary Fig. 6a). In contrast, OGDHL p.A400T compared to OGDHL WT was associated with significantly increased N-acetyl-aspartyl-glutamate, ribulose, and N-palmitoyltaurine; however these metabolites were not significantly elevated when comparing OGDHL mutant to all other groups (Supplementary Fig. 6b). Thus, while it's difficult to definitively comment on the new enzyme functionality, our data support a metabolic COMF for OGDHL p.A400T, in which the mutant may elicit an increase

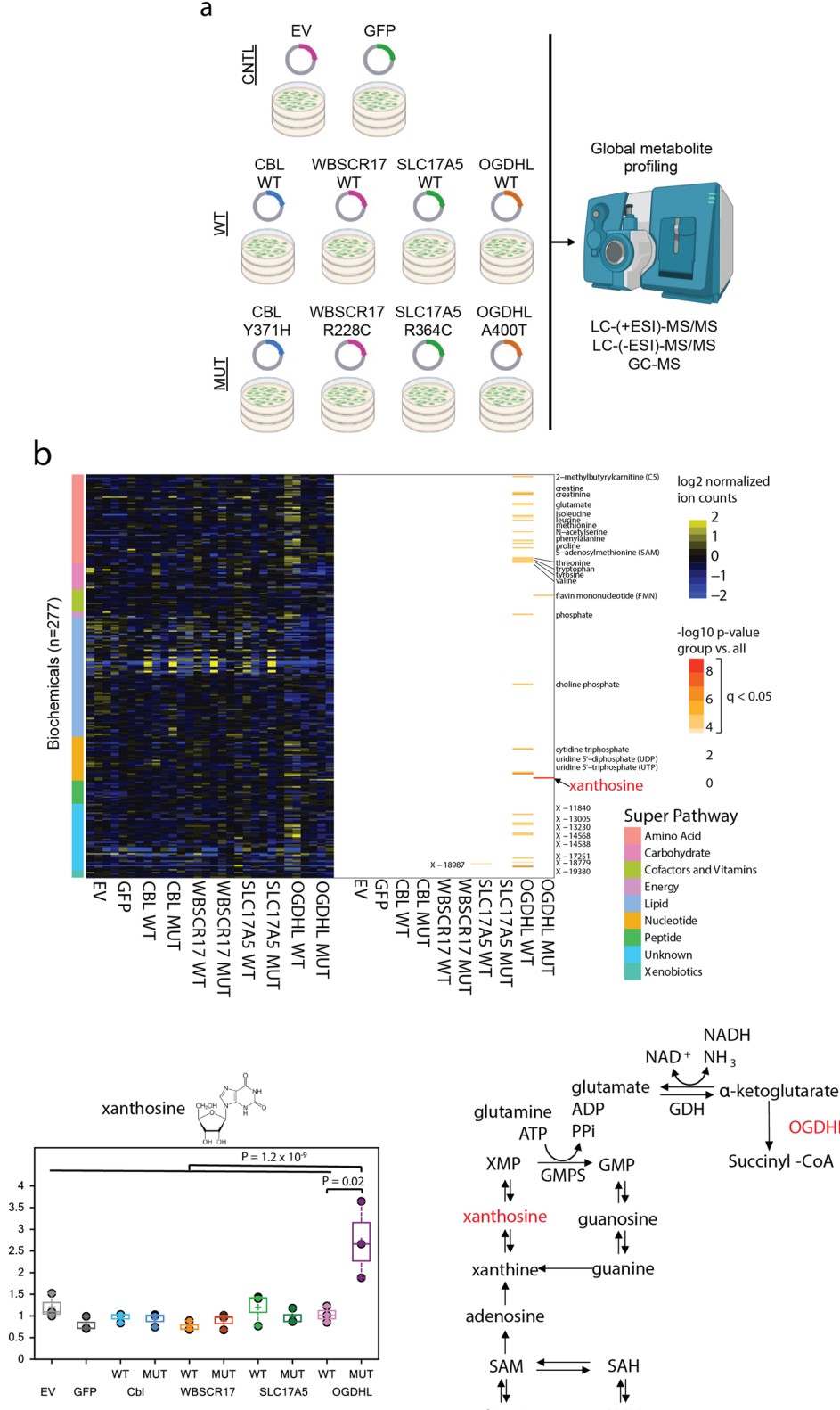

**Fig. 3 Global metabolite profiling to identify biochemicals perturbed by candidate mutants. a** Schematic of experiment showing cells transfected with indicated mutants, their analogous WT constructs, and empty vector (EV) or GFP controls. **b** Heat map showing 277 unique biochemicals as rows. Color key on left indicates biochemical super pathway. Heat map on left shows normalized ion counts for n = 3 independent transfection replicates for each construct (columns). Heat map on right shows biochemicals with a significant difference when compared between one group and all other groups (-log10 of q-value for Welch's t-test with Bonferroni correction). Biochemicals with q < 0.05 for any comparison are named on the right of the heat map. The biochemical with the most significant q-value named in red (xanthosine). Unique biochemicals of unknown structure are denoted by X-. **c** Normalized ion counts for xanthosine for n = 3 independent transfection replicates are shown on left with Welch's t test p-value shown for comparisons between OGDHL-MUT and OGDHL-WT and for OGDHL-MUT vs. all other samples. Metabolic pathway for purine degradation and S-adenosyl-methionine (SAM)-mediated xanthosine production shown on right.

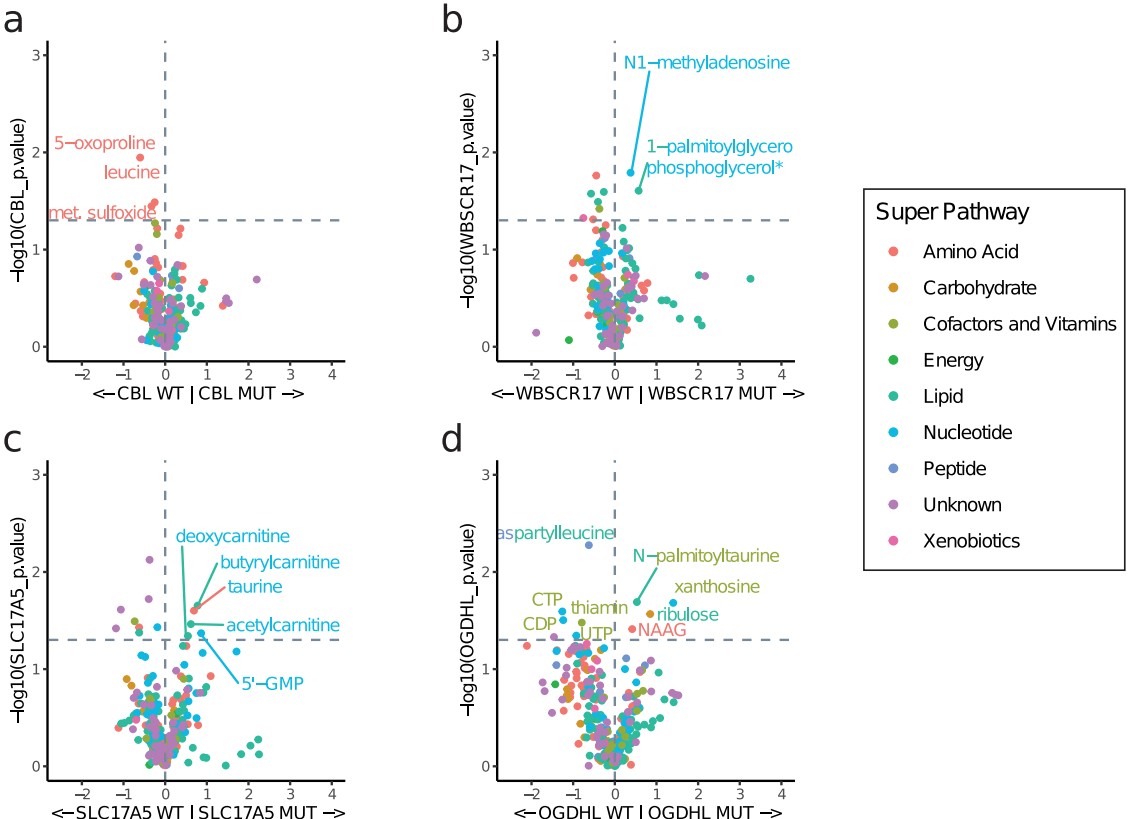

**Fig. 4 Comparisons between candidate COMF metabolomes and WT controls.** Volcano plots show fold-change (x-axis) and p-value (Welch's t-test, y-axis) for biochemical abundance in cells expressing each mutant (n = 3 biological replicates) compared to the respective wild type (WT) controls (n = 3 biological replicates). **a** CBL Y371H (MUT) vs. wild type (WT); **b** WBSCR17 (MUT) vs. wild type (WT)R228C; **c** SLC17A5 R364C (MUT) vs. wild type (WT); **d** OGDHL A400T (MUT) vs. wild type (WT). Biochemicals are colored based on metabolic super pathway. Significant metabolites increased in the mutant group (P < 0.05, Welch's t test, without FDR correction) and select metabolites increased in the WT groups are called out.

in xanthosine and/or other metabolites through downstream effects from a putative change in catalytic function or regulation of the enzyme.

**METIS2: Second generation pipeline to identify COMF cancer mutations.** We next sought to improve on the METIS1 approach. Since METIS1 was deployed in 2012, there have been dramatic increases in cancer genomic data and new in silico tools have emerged to model protein structures and characterize the pathogenicity of mutations. The increased data volume could be used to identify more COMF mutations in the long-tail distribution of cancer mutations.

We noted that 65X more missense mutations were identified in 2022 compared to 2012, from 2.1-fold more tumor samples (Fig. 5a). This led to an 308-fold increase in mutations observed 3 times or more, than our original recurrence threshold, making this threshold unviable (Fig. 5b). Our second-generation pipeline, METIS2, is outlined in Fig. 5c. METIS2 improves upon METIS1 by (i) using higher thresholds for recurrent mutations, (ii) incorporating FATHMM, a pathogenicity prediction tool, to filter out non-pathogenic mutations, and (iii) using AlphaFold to predict structures when protein structural data are unavailable (see Methods).

The expectations of COMF mutations were held constant with the addition of the assumption that they are generally pathogenic. We used the COSMIC v96 database of genome-wide and panel-based cancer sequencing studies. Based on the larger number of samples analyzed and baseline mutation rate, we estimated that

recurrence in 9+ unique patients' tumors and accounting for >25% of recurrent (9 + ) mutations in a gene would be extremely unlikely to occur by chance alone and would indicate positive selective pressure for the missense mutation (see Supplemental Note). We chose to use a FATHMM threshold score of −3, as this would minimize the number of false positives in the analysis (specificity of 0.99, sensitivity of 0.45)[27]. Examination of protein structure using available protein structure data or AlphaFold revealed 8 "hits" within 10 A of the predicted enzyme ligand binding site and 8 hits with the mutation located 10–15 A from the active site or putative active site (Supplementary Table 5). This structural analysis filtered 18 candidates that were distant from enzyme ligand-binding sites. Two germline polymorphisms were disregarded, leaving 6 potential COMF mutations.

METIS2 hits included OGDHLp.A400T (as in METIS1). We also identified 5 other candidate COMF mutations: an R283W mutation in D-Amino Acid Oxidase (DAO), which is commonly employed as industrial biocatalysts in the production of semi-synthetic cephalosporins and enantiomerically pure amino acids;[24] an L99F mutation in Microtubule Associated Monooxygenase 2 (MICAL2), a protein that has been shown to be a tumor-promoter;[28] two mutations, D638A and H639P, in Sphingomyelin Phosphodiesterase 3 (SMPD3); and A199P in TIMP Metallopeptidase Inhibitor 3 (TIMP3), a gene known for its potent tumor suppressive functions[29]. All of these mutations were recurrent, occurred at a conserved residue, and were located within the active site of the enzyme (Fig. 6a–l) .

Importantly, METIS2 identified OGDHLp.A400T as a COMF mutation candidate, consistent with the findings from METIS1

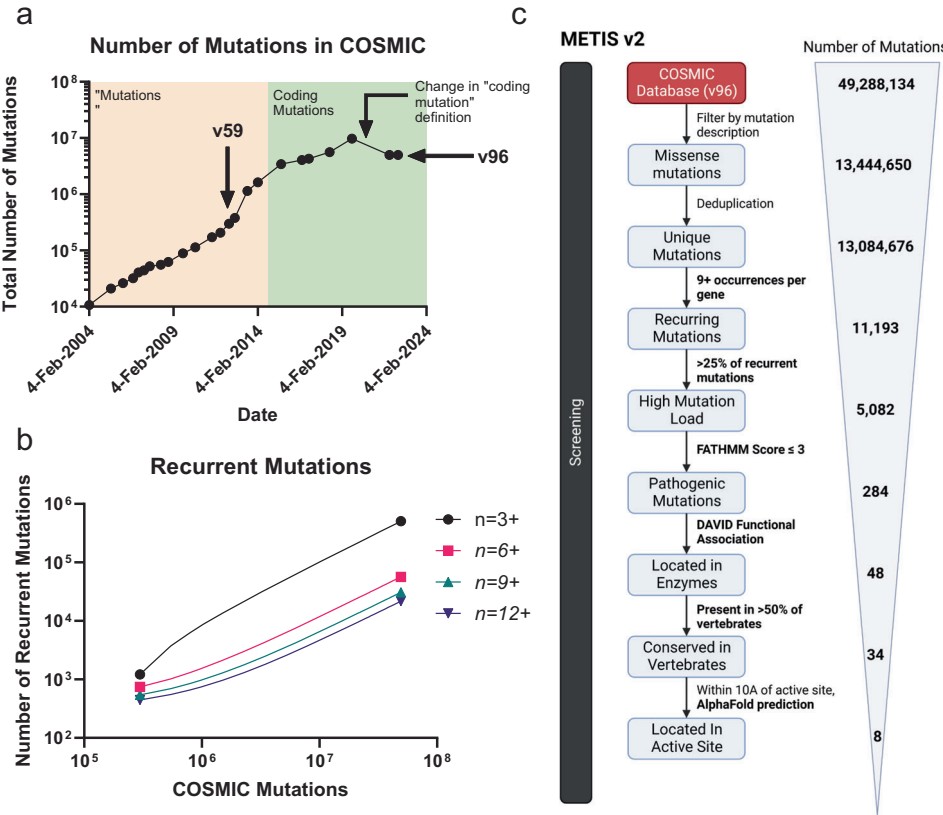

**Fig. 5 Second-generation pipeline to identify COMF mutations in cancer mutational data. a** Number of total missense mutations by year. **b** Number of 3 +, 4 +, … 9 +, 12+ recurrent mutations with COSMIC v59 (left) vs. COSMIC v96 (right) mutational data evaluated using METIS. An exponential Malthusian growth regression was used. **c** Schematic showing pipeline, with number of mutations filtered at each step.

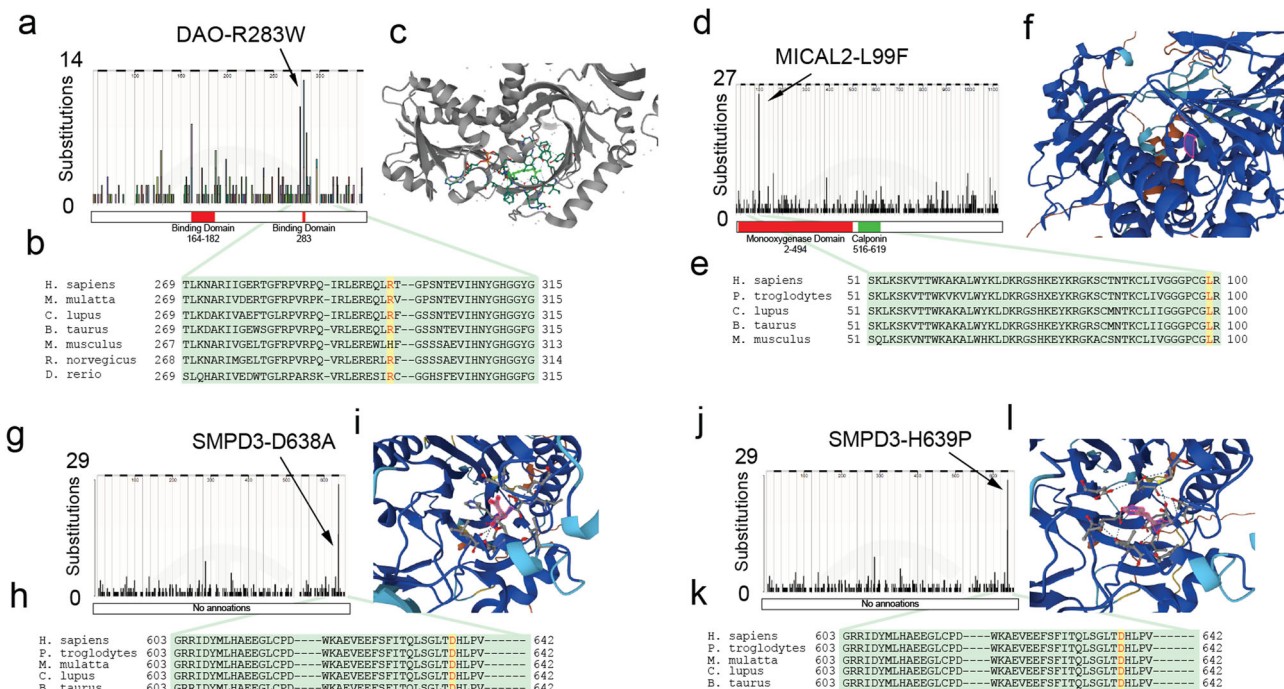

**Fig. 6 COMF enzymatic candidates nominated by METIS2. a** DAO mutational distribution in COSMIC v96 showing recurrent R283W mutations. **b** Multiple alignment with R283 shown in red. **c** Predicted structure of human DAO complex with R283 highlighted. **d–f** Same for MICAL2-L99F. **g–i** Same for SMPD3-D638A. **j–l** Same for SMPD3-H639P. Primary enzymatic function domains are represented by red boxes while secondary functions are represented by green boxes underneath the mutation plot. Enzymes without currently defined domains are annotated as such.

and the metabolomic screen. In the context of the additional 10 years of cancer mutation data and the implementation of more stringent filtering criteria, this finding supports the original filtering parameters in METIS1 and their continued use in METIS2. Moreover, as increased cancer mutation data is collected and stricter filtering parameters are introduced, we expect METIS to increase in its predictive power to accurately find COMF mutations.

This also may explain why METIS2 did not identify the other mutations identified by METIS1. The relatively few cancer mutations included in COSMIC ten years ago limited our ability to implement stricter parameters, which may have resulted in more false positives, as demonstrated by our metabolomic screen only confirming OGDHL A400T as a COMF (Supplementary Fig. 7a). In parallel, as METIS's predictive power increases, we also expect METIS to become more effective at finding COMF mutations. In particular, we expect METIS-like approaches to find a saturating number of "true" COMF mutations as available COSMIC mutational data increases and to level off at the total number of COMF, which previous studies have estimated 5% of mutations (Supplementary Fig. 7b).

## Discussion
Here we demonstrated an approach to mine cancer data for mutant proteins that can alter cellular metabolism, which provides new tools to enable novel metabolic engineering technologies. We applied a computational attempt at this approach to a cancer mutational dataset containing >200,000 missense mutations. A number of known and yet-unknown candidate COMF mutants were recovered. We generated a global metabolite profiling dataset for four of the novel candidate mutants. These data show that one or more of the candidates is uniquely perturbing the cellular metabolome due to a COMF. Recent datasets with exponentially more available cancer mutational data and updated analysis tools may provide more accurate COMF predictions. These results characterize a new approach to identifying novel COMF mutations, which may be useful to enable sustainable organic synthesis techniques and metabolic engineering processes. A graphical representation of our results can be found in Supplementary Fig. 7.

As expected, the majority of these predicted COMFs (77/125) primarily encoded transcription factors and signaling molecules well-known to be frequently mutated in cancer, such as GNAS, KRAS, and NRAS. The pipeline also identified several mutations in tumor suppressors such as TP53, NF1, and NF2 that likely represent particularly damaging missense mutations that inactivate the gene, rather than COMF mutations. Similarly, METIS candidates were also involved in tumor biology. For instance, OGDHL has been identified as a prognostic biomarker for hepatocellular carcinoma due to its role in reprogramming glutamine metabolism[26,30]. Further, CBLp.Y371H hyper-activates signaling downstream of hematopoietic growth factor receptors[31].

The novelty of integrating cancer genomics into metabolic engineering is promising, as has been demonstrated by other attempts to build predictors of GOF mutations. For instance, Coban-Akdemir et al. sought to identify potential GOF mutations, but this analysis was limited to mutations that cause premature termination codons, a relatively rare genomic event[29]. Shroff et al. used deep learning of protein structure to predict GOFs but could not link their relevance to human disease or larger metabolic pathway changes[32]. Liu et al. did not investigate mutations but rather individual amino acid positions within genes and their potential for GOFs. Unfortunately, it is difficult to draw direct comparisons between these methods and METIS as they were built for different purposes, with different data type

inputs, and with different types of mutations as outputs as outlined in Supplementary Table 6.

Compared to other state-of-the-art biocatalyst development techniques, such as directed evolution[33], METIS's strengths lie in its scalable, computational approach to identifying COMF mutations. Similarly, METIS will become increasingly high-throughput and accessible as publicly available cancer mutation data grows exponentially. METIS's filtering parameters are also highly customizable, which allows for purposeful screening that could be utilized for different biocatalyst development needs and even has implications for better understanding cancer metabolism. Towards this, METIS could easily be combined with other approaches to accelerate enzyme development to make up for its shortcomings: for example, METIS could suggest potential initial mutations for directed evolution to validate and build upon.

On the other hand, METIS also seems prone to type I errors, or the identification of false positives. This is evidenced by the candidate mutations in WBSCR17 and SLC17A5, which did not alter the metabolome of cells that were transfected with the same. This highlights the need for metabolic validation as well as the need for rich, abundant mutational data. False positives likely came about due to sparse data, as METIS1 was effective at predicting previously defined COMFs at "hotspot" mutations, such as IDH1 R132H. In fact, of its 48 hits, 34 were previously well-documented COMF mutations. In comparing METIS1 and METIS2, we also observed that, though there was an increase in the number of mutations that were filtered out by HomoloGene and protein structure analysis in METIS2, the ratio of these mutations to the number of recurrent mutations decreased from 2012 (Supplementary Fig. 7a). This likely is a result of the more stringent recurrence criteria set forth in METIS2, which perhaps decreased the likelihood of false positives in the screen. We expect METIS-like approaches to find a saturating number of "true" COMF mutations as available COSMIC mutational data increases and to level off at the total number of COMF, which previous studies have estimated 5% of mutations (Supplementary Fig. 7b). A method of avoiding false positives involves metabolic data, which may distinguish between COMFs and loss of function mutations. Considering that the original discovery of 2HG production by mutant IDH1 was found by noting a > 80-fold change in a metabolomic experiment, we posit that large effects would be more likely due to COMF than LOF[4].

The metabolomic data indicated that OGDHL A400T conferd a COMF that resulted in increased xanthosine production (2-fold increase). The fact the OGDHL-WT did not significantly alter xanthosine metabolism but the mutant did suggests that the COMF is a neomorphic function. While the direct substrates of mutant ODGHL (α-ketoglutarate and succinyl-CoA) were not detected within the screen, we did detect near-significant changes in several metabolites that are closely related to wild type OGDHL functions as shown in the Fig. 3c schematic. For instance, when comparing WT to mutant groups, glutamine was increased (Supplementary Fig. 8a, 4.3-fold, $p = 0.06$), which may reflect production of α-ketoglutarate; adenosine was increased (Supplementary Fig. 8b, 1.2-fold, $p = 0.06$); adenosine diphosphate was increased (Supplementary Fig. 8c, 2.6-fold, $p = 0.07$); and S-adenosylmethionine (SAM) was increased (Supplementary Fig. 8d, 1.7-fold, $p = 0.08$). In contrast, the OGDHL mutant appeared to completely lack the ability to cause the metabolic changes associated with OGDHL-WT (Supplementary Fig. 8a-d). This indicates that the OGDHL mutant completely loses wild type OGDHL activity, and may catalyze a novel reaction that either directly or indirectly results in xanthosine accumulation in cells. We have so far been unable to identify the substrates and products of a putative neomorphic OGDHL mutant reaction. Since OGDHL is an oxidoreductase, one possibility is that mutant

OGDHL could carry out an oxidoreductase reaction on a new substrate that could help generate ketone or other groups seen on xanthosine, or on a metabolite that ultimately is converted to xanthosine.

Purine alkaloids, which are derived from xanthosine, occur in and are largely derived from plants for industrial and medical applications. Purine alkaloids such as caffeine (coffee), theophylline (antiasthma drug), and theobromine (chocolate), play a significant role in pharmacology and food chemistry[34]. Traditional production of purine alkaloids relies on cultivation and compound extraction from plant biomass. These laborious strategies face long production times and intensive resource requirements from growing natural plant hosts. They not only have environmental implications but also result in a low or fluctuating supply of essential medicines[35]. Thus, the current production process of these important compounds are a significant barrier toward inexpensive and sustainable pharmaceutical development and commercial use. Furthermore, the chemical synthesis of xanthosine derivatives, in particular, is difficult due to the challenges in achieving selective alkylation or modification of each of the nitrogen atoms[36]. The desire to implement scalable, sustainable synthesis strategies has led to the desire to design metabolic engineering strategies to produce valuable plant-derived compounds. Our results suggest that METIS was able to identify COMF mutations, such as OGDHL.pA400T for the increased production of xanthosine. These mutations could be key to create relatively efficient cell-based synthesis strategies for key chemical compounds in a wide variety of chemical applications.

This work provides a framework for future efforts to comprehensively identify useful cancer-derived COMF mutations. We intentionally used extremely stringent filtering criteria and an early COSMIC database for these initial attempts at identifying cancer-derived COMF mutations, recovering <50 total candidates in these attempts. One group estimated that up to 5% of cancer mutations may be change of function mutations[17], raising the possibility that numerous cancer-derived COMF mutations may await discovery. Thus, future METIS efforts could use recent larger cancer mutation datasets, relaxed filtering thresholds, metabolic screen approaches with increased coverage, and improved statistical approaches to comprehensively identify a larger number of candidate cancer-derived COMF mutations. Confirmation in multiple cancer cell lines would be helpful to catch lineage-specific COMFs, such as CBL Y371H, which was identified in leukemia cells but not the HeLa cells used in this study[31]. Further optimization could also employ additional prediction methods to further estimate mutational effects on enzyme activity, such as MuPro and mahine learning approaches[37]. While the current work focused exclusively on missense mutations, future iterations of this approach could also be applied to identify fusions, deletions, or other structural alterations that may confer useful functions. Future work will also benefit from tools that were not widely available at the outset of this work. In particular, CRISPR can be used to introduce knockout mutations for metabolite profiling screens more faithfully and efficiently than our overexpression-based approach. Further, combinatorial CRISPR-Cas9 metabolic screens can allow high-throughput screening of COMF candidates[38]. This would greatly increase the rate at which METIS can confirm COMF mutations and identify the function that the mutation confers. Ultimately, future METIS improvements will increase the rate at which novel, useful catalysts can be developed using cancer mutational data.

## Methods

**The METIS bioinformatic pipeline.** COSMIC, v59, May 23, 2012 was used. R version 2.15.0 was used. The R code is supplied (Supplementary Code 1). Briefly, this code takes as input all coding mutations from a COSMIC database download. It filters for missense mutations, removes duplicate entries, and filters for mutations that occur in 3+ unique patient tumor samples, and those that account for >5% of all mutations in a given gene. A summary table is output. Analysis of the remaining candidate mutations is then carried out by excluding mutations as indicated in Supplementary Table 2. While not used for the current report, instructions to implement these filtering steps in Excel 2013 are also provided (Supplementary Code 2). HomoloGene was used to determine conservation. Proteins with available structures in the Protein Data Bank (PDB), or with homologous proteins in the PDB, were visualized using UCSF Chimera v1.5[39]. Possible germline mutations were identified using dbSNP and filtered out if MAF score > 0.01. PDB 2XTA was used for OGDHL, 1XHB for WBSCR17, and 1FBV for CBL, respectively. The statistical reasoning we used to determine recurrence cutoffs are described in the Supplementary Note. For comparative analyses, the number of missense and recurrent mutations were determined in the same manner as described above using COSMIC, v96.

**Plasmids.** cDNAs for CBL p.Y371H, WBSCR17 p.R228C, SLC17A5 p.R364C, OGDHL p.A400T were generated along with wild-type controls, GFP, or empty vector (see Supplementary Table 7 for sequences). For the OGDHL cDNAs, an open reading frame from OGDHL transcript variant 3 (NM_001143999.1) was used. This OGDHL cDNA was selected since it is shorter than other OGDHL transcript variants to facilitate cDNA transduction, but still contains the entire OGDHL enzyme domain of interest. In this OGDHL ORF, codon Ala191 is equivalent to the Ala400 codon associated with a candidate COMF mutation. SgfI/MluI restriction site ends were added by PCR to these cDNAs and cloned into pCMV6-Empty vector with C-terminal Myc-Flag epitope tags as described[5].

**Cell culture.** HeLa cells were obtained from ATCC. Cells were grown in DMEM media with 10% FBS and 1% Pen-Strep and split 3 times weekly using 0.05% trypsin. 10 cm were seeded with $5 \times 10^6$ cells, and 24 h later when plates reached approximately 80% confluency were transfected with 10 ug of each plasmid using Lipofectamine 2000 according to the manufacturer's instructions. Media was exchanged after 6 h at 37 °C in 5% CO$_2$. At 24 h post transfection, media was removed, cells were washed with 5 ml PBS, cells were immediately scraped into 1.7 ml tubes, pelleted at 3000 x g for 4 min, washed with 1 ml PBS, centrifuged at 3000 x g for 4 min, supernatant removed, and stored at −80 °C for further processing[5].

**Metabolomic profiling.** Metabolite profiling was carried out in collaboration with Metabolon as described previously[5]. The sample preparation process was carried out using the automated MicroLab STAR® system from Hamilton Company. Recovery standards were added prior to the first step in the extraction process for quality control purposes. Sample preparation was conducted using a proprietary series of organic and aqueous extractions to remove the protein fraction while allowing maximum recovery of small molecules. The resulting extract was divided into two fractions; one for analysis by liquid chromatography (LC) and one for analysis by gas chromatography (GC). Samples were placed briefly on a TurboVap® (Zymark) to remove the organic solvent. Each sample was then frozen and dried under a vacuum. Samples were then prepared for the appropriate instrument, either LC/mass spectroscopy (MS) or GC/MS.

The LC/MS portion of the platform was based on a Waters ACQUITY UPLC and a Thermo-Finnigan LTQ mass

spectrometer, which consisted of an electrospray ionization (ESI) source and linear ion-trap (LIT) mass analyzer. The sample extract was split into two aliquots, dried, then reconstituted in acidic or basic LC-compatible solvents, each of which contained 11 or more injection standards at fixed concentrations. One aliquot was analyzed using acidic positive ion optimized conditions and the other using basic negative ion optimized conditions in two independent injections using separate dedicated columns. Extracts reconstituted in acidic conditions were gradient eluted using water and methanol both containing 0.1% Formic acid, while the basic extracts, which also used water/methanol, contained 6.5 mM Ammonium Bicarbonate. The MS analysis alternated between MS and data-dependent MS2 scans using dynamic exclusion. For ions with counts greater than 2 million, an accurate mass measurement could be performed. Accurate mass measurements could be made on the parent ion as well as fragments. The typical mass error was less than 5 ppm. Ions with less than two million counts require a greater amount of effort to characterize. Fragmentation spectra (MS/MS) were typically generated in data dependent manner, but if necessary, targeted MS/MS could be employed, such as in the case of lower level signals.

The samples destined for GC/MS analysis were re-dried under vacuum desiccation for a minimum of 24 h prior to being derivatized under dried nitrogen using bistrimethyl-silyl-triflouroacetamide (BSTFA). The GC column was 5% phenyl and the temperature ramp is from 40° to 300 °C in a 16 min period. Samples were analyzed on a Thermo-Finnigan Trace DSQ fast-scanning single-quadrupole mass spectrometer using electron impact ionization. The instrument was tuned and calibrated for mass resolution and mass accuracy on a daily basis. The information output from the raw data files was automatically extracted as discussed below.

Compounds were identified by comparison to library entries of purified standards or recurrent unknown entities. Identification of known chemical entities was based on comparison to metabolomic library entries of purified standards. As of this writing, more than 1000 commercially available purified standard compounds had been acquired registered into LIMS for distribution to both the LC and GC platforms for determination of their analytical characteristics. The combination of chromatographic properties and mass spectra gave an indication of a match to the specific compound or an isobaric entity. Additional entities could be identified by virtue of their recurrent nature (both chromatographic and mass spectral). These compounds have the potential to be identified by future acquisition of a matching purified standard or by classical structural analysis.

**Immunoblots.** For each of the ten groups (four mutants, four corresponding WT controls, empty vector control, and GFP control), an additional 10 cm plate of HeLa cells was transfected in parallel with the cells used for global metabolite profiling. At 24 h post-transfection, this parallel plate was washed twice with PBS, 600 ul of lysis/loading buffer with protease inhibitors was added as described previously[8], and cells were scraped into 1.7 ml tube, and incubated at 4 °C overnight. 20ul of each lysate was then added to 20 ul of Laemmli sample buffer (BioRad) containing 5% b-mercaptoethanol, boiled at 100 °C for 5 min, centrifuged 10 min 13,000 rpm at 4 °C and the supernatants were loaded onto a 4–12% Tris-glycine gel. Immunoblots were carried out using anti-FLAG (1:1000, TA100011, OriGene) as described previously[40]. To optimize immunoblots for membrane-associated proteins in SLC17A5-expressing cell samples, the same procedure was carried out without a boiling step. Instead, the samples were sonicated for 5 min in 30-s on, 30-s off pulses prior to adding 1X Laemmli sample buffer.

**Sensitivity Analysis.** To roughly assess the accuracy of our METIS1 pipeline, we selected n = 42 gold-standard COMF mutations (Supplementary Table 4). These mutations were all well-known COMF mutations, including IDH1p.R132H and EZH2p.Y646F. We conservatively estimated a priori the presence of approximately 300 enzymes involved in cancer-related metabolic pathways. True positives were mutations that existed both on the gold-standard list and the list of predicted mutations. False positives were mutations that were loss of function tumor suppressors (such as TET2p.I1873T). True negatives were mutations not identified as COMFs that fell into the 258 mutations/genes that were not included in the gold-standard panel. False positives were mutations that were predicted by COMFs but did not fall into the gold-standard panel.

**Statistics and Reproducibility.** For pair-wise comparisons of metabolomic data (WT vs. mutant, and in-group vs. out-group), Welch's t-tests were used with Bonferroni correction for multiple hypothesis testing. For multi group statistical designs repeated measures ANOVA was used. Statistical analyses are performed with the program "R" http://cran.r-project.org/.

## Data availability
The mutation and metabolomic datasets generated during and/or analyzed during the current study are available in the supplemental materials and COSMIC database (https://cancer.sanger.ac.uk/cosmic). The metabolomic datasets generated during and/or analyzed during the current study are also available on Figshare (https://doi.org/10.6084/m9.figshare.21293739) or from the corresponding author on reasonable request.

## Code availability
Code used in this manuscript is available in the supplementary information and was also deposited into Figshare (https://doi.org/10.6084/m9.figshare.21293739).

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

## Acknowledgements

We thank Hai Yan and David G. Kirsch for critical feedback on the project and manuscript. We thank Weixin Zhou, Paula Greer, and Robert Mohney for technical assistance. The work was supported by institutional funds to Z.J.R., and Z.J.R. is supported by career development funds from a K08CA2560450, the Pediatric Brain Tumor Foundation, St. Baldrick's Foundation, Emily Beazley's Kures for Kids, ChadTough Defeat DIPG. K.J.T. was supported by a scholarship from the Amgen Foundation and a Banneker/Key Scholarship.

## Author contributions

Z.J.R. and K.J.T. conceived the study, designed experiments, and wrote the manuscript. Z.J.R, K.J.T, J.A.R, B.H.D, M.S.W, and C.J.P performed analysis and molecular biology work. J.A.R., B.H.D., M.S.W., and C.J.P. provided revisions to the manuscript.

## Competing interests

The authors declare the following competing interests: ZJR holds patents that are managed by the Office of Licensing and Ventures at Duke University that relate to cancer-derived enzyme redesign (US8691960B2). The other authors declare no competing interests.
