## [Peer review file · Communications Biology]

Reviewers' comments:

Reviewer #1 (Remarks to the Author):

In this manuscript, the author developed a bioinformatics pipeline to screen cancer sequencing data across the genome for potentially cancerous COMF mutations. The author then generated a metabolomics dataset to identify any metabolic disruptions that may have been caused by the top four identified COMF mutants. The dataset identified cell metabolites uniquely perturbed by the candidate COMF mutants but not by their wild-type controls, indicating that these mutants conferred an impact on the cellular metabolome. This suggests that the presence of these mutants has an impact on the cell's metabolome. The present study proposes a workflow for identifying potential COMF mutants from cancer mutation profiles, provides a compilation of nominated candidates and demonstrates an unbiased metabolomics analysis for identifying mutant function. However, we have following comments on the evaluation of result section. The study also needs more systematic evaluations, particular to evaluate the correctness of the COMF pipeline.

Major:

1. In the "METIS1: a pipeline to identify COMF cancer mutations" section of the results, the author mentions the criteria for selecting mutations that occurred in ≥ 3 patients and accounted for $>5\%$. How were these thresholds defined? In addition, the frequency distribution of mutations in cancer genomes follows a long-tail distribution, and using these filtering conditions would eliminate the vast majority of mutations (only 735 mutations were selected from 207,816 mutations). Therefore, would this step result in fewer COMF mutations being identified in the subsequent steps?
2. Should a step be added to the METIS1 pipeline to filter driver mutations, to ensure that the identified COMF mutations have significant biological functions in cancer?
3. The author is suggested to collect previously verified COMF mutations as a gold standard to validate the accuracy of the METIS1 pipeline, and calculate the sensitivity and specificity of the predicted results.
4. In the "Unbiased global metabolite profiling of candidate COMF mutants" section of the results, the METIS1 pipeline identified four candidate COMF mutations, but only OGDHL p.A400T was experimentally validated to cause changes in metabolites. Thus, the predictive model has a low success rate of 25%. How can this be explained?
5. In the "Comparison of first- and second-generation METIS pipelines" section of the results, the study shows that the predictive results of METIS2 are better than those of METIS1. If METIS1 and METIS2 have opposite predictions for the same gene, how should the predictive results for that gene be explained? Please also show a list of genes with contradictory predictions.
6. Does the METIS pipeline's identification of COMF mutations depend on the sample size and the number of mutations annotated in COSMIC? Would a larger sample size or a greater number of mutations result in the identification of more predicted COMF mutations? If the predictive results for COMF mutations are influenced by these factors, it may indicate that the predictive model's results are unstable.
7. The author can collect and utilize a gold standard dataset of COMF mutations to compare the accuracy of the METIS method with other methods for identifying COMF mutations, rather than simply discussing the shortcomings of other methods in identifying COMF mutations.
8. The author needs to investigate the important biological implications of the identified COMF mutations, such as whether they play a role in cancer development or can predict patients' response to drugs.

Minor:

1. The author needs to provide information on the mutation sites filtered at each step of the METIS pipeline.

Reviewer #2 (Remarks to the Author):

Tu et al describe a framework for identifying metabolic enzymes with altered catalytic function in large-scale mutation data from cancer patients. The idea is interesting and certainly offers a distinct and new view on the potential in mining genomic resources/molecular data generated with

modern profiling technologies.

The authors have used different datasets and generated testable predictions of mutated enzymes with new functions. While the filter criteria per se seem to be robust, a few questions regarding the information content of the predictions remain and should be addressed.

Major comments:

> In the abstract and introduction sections, the authors emphasize the need to "enable new organic synthesis techniques" and "to create new biocatalysis for a variety of fields". Their approach aims at uncovering mutations that "confer useful new functions to enzymes". This is an interesting idea, however the authors do not really explain what defines "useful", or in what way their filter criteria concretely search for "usefulness"? To put it bluntly, just catalyzing a different reaction that normally would not immediately make a mutant enzyme's function "useful". It would be of interest to get more insight into the author's view on this.

> The authors present a test case (OGDHL) where metabolic profiling is used to assess the result of altered enzyme activity in mutants. However, the experiment was done in the complex setting of native metabolism in HeLa cells, so I think the interpretation is not so trivial. The metabolic profiling data shows significant changes associated with the mutated enzyme, which the authors ascribe to altered functionality. However, it is more likely that what we see are already downstream metabolic effects that have arisen from changes in the upstream reaction catalyzed by OGDHL. From global metabolite profiling in cell extracts and without additional data, I don't think it is possible to conclude on the new enzyme functionality (i.e. substrate-product conversion). In terms of an application, that potentially means if one wanted to use this enzyme to generate xanthosine, it would always have to be in a cell-based system.

> On a related note, since the authors indicate repeatedly that their predicted enzymes could be used as catalysts: To use a given candidate as a biocatalyst, one would need to know at least the substrate and product of the catalyzed reaction, if not information on the reaction mechanism. While some of these aspects are likely outside the scope of this bioinformatics-focused manuscript, it seems that the authors should be able to partially address them with the available metabolic profiling data. For example, I was surprised that no comment was made on the substrate and product of the "original" OGDHL reaction. Were the changes not significant, or were the metabolites not detected at all? Or do the authors think that substrate and product are completely different from the WT enzyme, and if so why and which do they think are the new reactants? If a limited coverage of metabolism by the metabolic profiling technology used (here: Metabolon company), this should be addressed in the discussion.

> In lines 387-388 the authors indicate that metabolite profiling can be used to differentiate missense from loss of function. I'm not sure whether this is systematically possible, especially if substrates and products may not always be detected. I would kindly ask the authors to clarify which type of metabolic changes would indicate a LOF vs. a COMF mutation and the interpretation of metabolic changes in general (see also comments above).

Minor comments:

Line 300/301 65X, 2.1X use "-fold" or "-times"

Line 310/311 word repetition of "additional" in the same sentence

Lines 309-324 seems quite hypothetical/speculative, perhaps is better suited in the discussion section rather than the results.

Line 339/340 unclear wording "the novelty [...] is promising [...]"

357-359: Please clarify more explicitly in what way METIS is not accurate, how this was assessed and what the potential reasons and therefore solutions could be. Also the following sentence "METIS could easily be combined with other approaches to make up for its shortcomings." - please be explicit, what approaches, how, etc.

Comments from Reviewer 1

Referee: 1. In the "METIS1: a pipeline to identify COMF cancer mutations" section of the results, the author mentions the criteria for selecting mutations that occurred in ≥ 3 patients and accounted for $>5\%$. How were these thresholds defined? In addition, the frequency distribution of mutations in cancer genomes follows a long-tail distribution, and using these filtering conditions would eliminate the vast majority of mutations (only 735 mutations were selected from 207,816 mutations). Therefore, would this step result in fewer COMF mutations being identified in the subsequent steps?

Author: Thank you for your questions and comments. The thresholds for selecting recurrent mutations were outlined in the supplemental note. Briefly: we estimated that there was a possibility of 7.6×10^{-8} that a missense mutation was located in a codon. We then used a battery of false discovery rate tests, including BH, Holm, and Benjamini-Yekutieli, which were calculated using R. We set $\text{adj}P < 0.0001$ for all FDR tests, and 3+ occurrences were determined to be significant for a gene to be recurrent. Here, we opted to use a strict significance criteria as a proof of concept, but this selection criterion can be loosened/tightened in future iterations of the pipeline. We now refer readers to the method used to select our METIS1 thresholds in the first part of the results (please see page 7, paragraph 3).

We agree that the long-tail distribution of cancer mutations makes identifying "rare" COMF mutations difficult. Future systematic analyses and bigger datasets could be used to identify mutations in the "tail" of the distribution. This was part of the reason why we completed METIS2: to use the substantially larger dataset that is currently available to identify potential mutations that were filtered out in the original METIS1 analysis. We now explain this rationale for using larger datasets in METIS2 in that section of the results (please see page 13, paragraph 2).

Referee: 2. Should a step be added to the METIS1 pipeline to filter driver mutations, to ensure that the identified COMF mutations have significant biological functions in cancer?

Author: Thank you for your thoughtful suggestion. To address this concern, we have incorporated a final step in the METIS1 pipeline (please see page 9, paragraph 2). Following the mutation identification process, we now conduct a rational literature review to carefully assess the biological functions of the COMF mutations in the context of cancer. This additional step allows us to identify any non-significant mutations and focus on those with relevant biological implications. It should be noted that some identified COMF mutations are currently not well-studied, and lack of literature may not necessarily mean the absence of a COMF.

Referee: 3. The author is suggested to collect previously verified COMF mutations as a gold standard to validate the accuracy of the METIS1 pipeline, and calculate the sensitivity and specificity of the predicted results.

Author: Thank you for your comment. To estimate the accuracy of our METIS1 pipeline, we selected $n=42$ gold-standard COMF mutations. These mutations were all "infamous" COMF mutations, including IDH1p.R132H and EZH2p.Y646F (please see new **Table S4**). We conservatively estimated *a priori* the presence of approximately 300 enzymes involved in

cancer-related metabolic pathways. True positives were mutations that existed both on the gold-standard list and the list of predicted mutations. False positives were mutations that were loss of function mutations in tumor suppressors (such as TET2p.I1873T). True negatives were mutations not identified as COMFs that fell into the 258 mutations/genes that were not included in the gold-standard panel. False positives were mutations that were predicted by COMFs but did not fall into the gold-standard panel.

Using this framework, we estimated the sensitivity of our analysis as 86% given its ability to detect most (36/42) known COMF mutations (please see **Supplementary Table S2 in manuscript**). We estimated the specificity at 95% (246/260) given the high number of yet-unestablished COMF mutations and enzymes involved in metabolic pathways. Overall accuracy was estimated at 93% (282/300). Though there were some limitations in this method, such as there being no way to know *a priori* which COMFs were actual COMFs, this estimate shows that METIS1 was still reasonably effective at identifying known COMF mutations.

We now summarize these estimates in the Results (please see page 8, paragraph 2) and include details for how we identified the gold standard list and performed the sensitivity, specificity, and accuracy analyses in the **Supplementary Note**.

Referee: 4. In the "Unbiased global metabolite profiling of candidate COMF mutants" section of the results, the METIS1 pipeline identified four candidate COMF mutations, but only OGDHL p.A400T was experimentally validated to cause changes in metabolites. Thus, the predictive model has a low success rate of 25%. How can this be explained?

Author: Thank you for your insightful comment. We appreciate your attention to the filtering process and its impact on the predictive model's success rate. We filtered out previously identified COMF mutations from METIS1, including the IDH1 R132H mutation, to demonstrate the novelty of our findings. Taking these into account, the predictive model achieved an accuracy of approximately 93% (see above), indicating the model's robustness and efficacy in identifying novel COMF mutations. We have made a note in the results section clarifying the fact we filtered out previously identified COMFs (page 8, paragraph 2).

Recent studies have shown that CBL, which was nominated by METIS1 was also, in fact, a gain-of-function mutation. This difference has been modified within the manuscript. Regarding the GALNT17 and SLC17A5 mutations, which did not demonstrate significant metabolic changes, the relatively low amount of available mutational data on these novel mutations may have influenced their correlation with predicted COMF changes.

Referee: 5. In the "Comparison of first- and second-generation METIS pipelines" section of the results, the study shows that the predictive results of METIS2 are better than those of METIS1. If METIS1 and METIS2 have opposite predictions for the same gene, how should the predictive results for that gene be explained? Please also show a list of genes with contradictory predictions.

Author: METIS1 and METIS2 only contained one contradictory prediction for the same gene: METIS1 predicted a COMF in KDR T771R, whereas METIS2 predicted a COMF in KDR Q472H (**Figure 1 below**).

The difference in findings may be attributed to differences in the filtering criteria set forth between METIS1 and METIS2. For example, METIS1 only required a prediction to take up 5% of mutations in a gene, whereas METIS2 required mutations to make up at least 20%. The difference in findings may also be attributed to data availability. The relatively little data available during the execution of METIS1 (in 2013) likely have identified mutations that with the accumulation of additional sequencing data were found to not actually be highly recurrent. The data available at the time METIS2 was executed (in early 2023) improved the resolution of actual recurrent mutations, allowing mutations like Q472H to be selected.

Referee: 6. Does the METIS pipeline's identification of COMF mutations depend on the sample size and the number of mutations annotated in COSMIC? Would a larger sample size or a greater number of mutations result in the identification of more predicted COMF mutations? If the predictive results for COMF mutations are influenced by these factors, it may indicate that the predictive model's results are unstable.

Author: Thank you for the insightful comment. METIS's identification of COMF mutations depends on the stringency of the filtering criteria imposed upon it. More stringent filtering criteria will predict fewer COMF mutations. If the filtering criteria were kept at the same relative stringency (based on adjusted significance on a binomial distribution), then a larger sample size or a greater number of mutations should not result in the identification of more predicted COMF mutations.

Referee: 7. The author can collect and utilize a gold standard dataset of COMF mutations to compare the accuracy of the METIS method with other methods for identifying COMF mutations, rather than simply discussing the shortcomings of other methods in identifying COMF mutations.

Author: Thank you for the comment. While we agree that comparing different methods is important, it would be difficult to do so here given the novelty of the pipeline and the heterogeneity between current computational predictors. The discussed computational methods were built for different purposes and upon considerably different datasets, making benchmarking these methods impossible (**Table 1** below, or please see new **Supplementary Table 9** in the revised manuscript). For example, the output for the pipeline built by Coban-Akdemir, 2018 was frameshifting indels while the output for METIS were missense mutations. Thus the same gold-standard dataset would not be applicable to both. It thus would be difficult to draw meaningful quantitative comparisons between the different methods. Hence, we were unable to include such an analysis within the manuscript. To highlight these differences, we now discuss these differences in the Discussion (please see page 18, paragraph 1) and provide a table comparing the disparate inputs and outputs of the different methods (**Table 1** below, or new **Supplementary Table 9**).

Referee: 8. The author needs to investigate the important biological implications of the identified COMF mutations, such as whether they play a role in cancer development or can predict patients' response to drugs.

Author: Thank you for your comment. To address this concern, we now provide an explanation of the roles of OGDHL and CBL in cancer development within the discussion section as follows:

“METIS candidates were also involved in tumor biology. For instance, OGDHL has been identified as a prognostic biomarker for hepatocellular carcinoma due to its role in reprogramming glutamine metabolism. Further, CBLp.Y371H hyper-activates signaling downstream of hematopoietic growth factor receptors.” (page 17, paragraph 2)

Referee: Minor comment: 1. The author needs to provide information on the mutation sites filtered at each step of the METIS pipeline.

Author: Given the large number of data points included in each step of the filtering process, it would be difficult to include all mutation sites filtered at each step within a supplemental table (such a table would include hundreds of thousands of entries to include each individual point mutation). We attempted to summarize the mutants filtered at each step in **Table S1** and in **Figures 1 and 5**. Further, we have made it relatively simple for researchers to duplicate our code to explore the data themselves in either R or Excel following the instructions in the Supplementary code. We hope that this will be sufficient in helping researchers in accessing the information on the mutation sites filtered at each step of the METIS pipeline. We have clarified in the text how to more access these details within the results section (page 8, paragraph 1) as follows:

“After this analysis, 735 mutations were identified in 125 genes (details on filtered out mutation sites in Supplementary Table 1).”

Comments from Reviewer 2

Referee: “Tu et al describe a framework for identifying metabolic enzymes with altered catalytic function in large-scale mutation data from cancer patients. The idea is interesting and certainly offers a distinct and new view on the potential in mining genomic resources/molecular data generated with modern profiling technologies.

The authors have used different datasets and generated testable predictions of mutated enzymes with new functions. While the filter criteria per se seem to be robust, a few questions regarding the information content of the predictions remain and should be addressed.

Major comments:

> In the abstract and introduction sections, the authors emphasize the need to "enable new organic synthesis techniques" and "to create new biocatalysis for a variety of fields". Their approach aims at uncovering mutations that "confer useful new functions to enzymes". This is an interesting idea, however the authors do not really explain what defines "useful", or in what way their filter criteria concretely search for "usefulness"? To put it bluntly, just catalyzing a different reaction that normally would not immediately make a mutant enzyme's function "useful". It would be of interest to get more insight into the author's view on this.”

Author: Thank you for this question; we agree that the description of our goals was vague in the original manuscript. Mutated enzymes were considered useful if they improved synthetic route efficiency in terms of the number of steps and/or the yields of these steps or introduced a more sustainable production method. We previously demonstrated a proof of concept in a previous paper (Reitman et al., 2012, Nat Chem Biol), where a R132H mutation in IDH1 was used to

develop an improved synthetic route for adipic acid, a precursor for nylon. We have now made this distinction clearer within the introduction of the manuscript (page 5, paragraph 2).

Referee: “The authors present a test case (OGDHL) where metabolic profiling is used to assess the result of altered enzyme activity in mutants. However, the experiment was done in the complex setting of native metabolism in HeLa cells, so I think the interpretation is not so trivial. The metabolic profiling data shows significant changes associated with the mutated enzyme, which the authors ascribe to altered functionality. However, it is more likely that what we see are already downstream metabolic effects that have arisen from changes in the upstream reaction catalyzed by OGDHL. From global metabolite profiling in cell extracts and without additional data, I don't think it is possible to conclude on the new enzyme functionality (i.e. substrate-product conversion). In terms of an application, that potentially means if one wanted to use this enzyme to generate xanthosine, it would always have to be in a cell-based system.”

Author: We agree with the reviewer that in terms of application, our results would be most applicable in a cell-based system. We believe that this may still potentially be an improvement on current xanthosine production methods, which relies heavily on plant biomass and intense agricultural resources. We now make this point more clearly within the Discussion section of the manuscript (page 21, paragraph 1).

We also agree with the reviewer that it is not possible to definitively conclude on the new OGDHL enzyme functionality. Please see our response to the next reviewer question below regarding new changes to the manuscript that address this point.

Referee: “On a related note, since the authors indicate repeatedly that their predicted enzymes could be used as catalysts: To use a given candidate as a bioacatalyst, one would need to know at least the substrate and product of the catalyzed reaction, if not information on the reaction mechanism. While some of these aspects are likely outside the scope of this bioinformatics-focused manuscript, it seems that the authors should be able to partially address them with the available metabolic profiling data. For example, I was surprised that no comment was made on the substrate and product of the "original" OGDHL reaction. Were the changes not significant, or were the metabolites not detected at all? Or do the authors think that substrate and product are completely different from the WT enzyme, and if so why and which do they think are the new reactants? If a limited coverage of metabolism by the metabolic profiling technology used (here: Metabolon company), this should be addressed in the discussion.”

Author: Thank you for your comment. We agree that a limitation of the study is that the substrate and product of the novel catalyzed reaction is unknown. Our metabolic screen data may suggest the following regarding a change in function of the OGDHL enzyme:

To get a better idea of how the OGDHL function may be altered, we tried observing the abundance of the substrate and product of the original WT OGDHL reaction. Unfortunately, our metabolite screen did not include the direct substrate of OGDHL WT (2-oxoglutarate). We have also noted that our metabolic screen had a limited coverage of metabolism within our discussion as another limitation (page 21, paragraph 2). However, our metabolic screen did include biochemicals upstream (citrate) and downstream (malate) of the OGDHL putative metabolic

pathway. Both metabolites were not significantly changed in cells transfected with OGDHL A400T (as compared with OGDHL-WT, with fold change of citrate 0.91, $P=0.84$, and fold change of malate 0.72, $P=0.33$). This lack of significant changes may have been due to the fact that OGDH, a paralog of OGDHL, was maintaining normal catalytic output. While the direct substrates of OGDHL were not detected, we did detect near-significant changes in several metabolites related to wild type OGDHL function. For instance, when comparing WT to mutant groups, glutamine was increased (4.3-fold, $p=0.0576$), which may reflect production of α -ketoglutarate; adenosine was increased (1.2-fold, $p=0.0608$); adenosine diphosphate was increased (2.63-fold, $p=0.0652$); and S-adenosylmethionine (SAM) was increased (1.7-fold, $p=0.075$). We now mention this within the discussion section (page 20, paragraph 1) and **Extended Data 7**.

As the reviewer mentioned, the increase in xanthosine may be a result of downstream effects of the OGDHL mutation. It is possible that the “new” product of mutated OGDHL may have spurred xanthosine production as opposed to directly catalyzing its creation, which is why we now use more speculative language when describing this result. As the reviewer mentions, it is not possible to definitively comment on the new enzyme functionality. We have made this distinction within the Discussion section and toned down our claims (please see page 19, paragraph 2).

Nevertheless, we may be able to derive some insight into how the xanthosine was produced after OGDHL was mutated. There are 4 main pathways of xanthosine production: from de novo purine synthesis (IMP), from the adenine nucleotide pool (AMP), from the guanine nucleotide pool (GMP), and from the S-adenosyl-methionine (SAM) cycle. While we did not notice changes in the IMP, AMP, or GMP routes, we did notice near-significant changes in SAM cycle metabolites. Our results raise the possibility that mutant OGDHL sped up the production of xanthosine through perturbing the SAM cycle. To summarize this observation conservatively, we have now added a brief comment on this hypothesis in the discussion (page 20, paragraph 1).

Referee: “In lines 387-388 the authors indicate that metabolite profiling can be used to differentiate missense from loss of function. I'm not sure whether this is systematically possible, especially if substrates and products may not always be detected. I would kindly ask the authors to clarify which type of metabolic changes would indicate a LOF vs. a COMF mutation and the interpretation of metabolic changes in general (see also comments above).”

Author: Thank you for the comment. To distinguish between LOF and COMF mutations, we can consider that the original discovery of 2HG production by mutant IDH1 was found by noting a large fold-change of 2HG in an unbiased metabolomic experiment (>80-fold change, ref: Dang et al., 2009). Similarly, we found a relatively large FC in xanthosine (>2-fold) within our experiment. Thus, we posit that large metabolic effects would be more likely due to COMF than LOF mutations and can serve as an identifier for COMF mutations (page 18, paragraph 1).

Referee: “Minor comments: Line 300/301 65X, 2.1X use “-fold” or “-times””

Author: Thank you for your comment, we’ve incorporated this change.

Referee: “Line 310/311 word repetition of “additional” in the same sentence”

DukeHealth

Department of Radiation Oncology

THE
PRESTON ROBERT TISCH
BRAIN TUMOR CENTER
at Duke...there is HOPE

Author: Thank you for your comment, we've incorporated this change.

Referee: "Lines 309-324 seems quite hypothetical/speculative, perhaps is better suited in the discussion section rather than the results."

Author: Thank you for your comment, we've moved this section to the discussion section. We have also moved part of the corresponding figures to **Figure 5** and **Extended Data 6**.

Referee: "Line 339/340 unclear wording "the novelty [...] is promising [...]"

Author: Thank you for your comment, we've incorporated this change.

Referee: "357-359: Please clarify more explicitly in what way METIS is not accurate, how this was assessed and what the potential reasons and therefore solutions could be. Also the following sentence "METIS could easily be combined with other approaches to make up for its shortcomings." - please be explicit, what approaches, how, etc."

Author: Thank you for your comment, we've incorporated this change.

Figure 1. Mutational distribution of gene KDR. Red arrows indicate METIS2 (Q472H) and METIS1 (T771R) hits, respectively.

Characteristic	METIS	Coban-Akdemir, 2018	Shroff, 2020	Liu, 2015
Foundation Data	Catalogue Of Somatic Mutations In Cancer (COSMIC)	Atherosclerosis Risk in Communities Study (ARIC), Exome Aggregation Consortium (ExAC), Baylor-Center for Mendelian Genomics.	Protein Data Bank	Thyroid Stimulating Hormone Receptor Mutation Database II, IARC TP53
Mutation identified	Change of metabolic function	Gain of function	Gain of function	Loss of function, gain of function, switch of function, conservation of function
Computational Method	Recursive screening	Nonsense Mediated Decay Escape Intolerance Score calculator	Deep Learning	Hidden Markov model
Predictors	Recurrence, enzyme structural analysis, conversation, pathogenicity	Transcript sequence, frameshift site, premature termination codon efficiency	Protein structure	Conversation, pathogenicity
Metabolomic analysis?	Yes	No	No	No

Output	Missense mutations	Frameshifting indel	Missense mutations	Point locations in genes
Organism	Homo sapiens	Homo sapiens	Escherichia coli	Homo sapiens

Table 1. Comparisons between mutation function predictors.

REVIEWERS' COMMENTS:

Reviewer #1 (Remarks to the Author):

The authors have appropriately responded to the previous comments. However, one remaining concern pertains to the experimental validation in HeLa cells. Specifically, while the predictive model has shown a commendable accuracy of 93% when using gold standard data, why is there a substantial discrepancy with the success rate in HeLa cells, which stands at merely 25%?

Reviewer #2 (Remarks to the Author):

I thank the authors for considering all comments and critically assessing their implications for the manuscript. All points have been addressed, and the authors have now added adequate comments on several critical aspects of the interpretation of their data. I support publication of this paper in this revised version.

DukeHealth

Department of Radiation Oncology

THE
PRESTON ROBERT TISCH
BRAIN TUMOR CENTER
at Duke...there is HOPE

October 4, 2023

Dr. Christina Karlsson Rosenthal, PhD
Chief Editor, Communications Biology
4 Crinan Street
London N1 9XW, UK

Zachary J. Reitman, MD, PhD

Assistant Professor of Radiation Oncology,
Pathology, and Neurosurgery
Attending Physician: CNS Radiation Oncology
The Preston Robert Tisch Brain Tumor Center
Duke University School of Medicine

RE: Manuscript Resubmission

Dear Dr. Christina Rosenthal,

We write to submit a revised version of our manuscript for your consideration, “Mining Cancer Genomes for Change-of-Metabolic-Function Mutations to Design Novel Catalysts” (COMMSBIO-23-1326-T). The reviewers and editor suggested acceptance of the manuscript with minor revisions. We thank the reviewers for their second round of revisions, which has allowed us to improve the manuscript slightly. We feel that we have been able to address all of the reviewers’ concerns. In our rebuttal letter, we describe the changes we made to the work and respond to the comments from the reviewers at Communications Biology.

We appreciate your continued consideration of our manuscript for publication. I am able to answer any questions that may arise. We look forward to hearing from you.

Kind Regards,

Zachary J. Reitman, MD, PhD (he/him/his)
Assistant Professor of Radiation Oncology, Pathology, and Neurosurgery
Attending Physician: CNS Radiation Oncology
ChadTough Defeat DIPG New Investigator
Emily Beazley’s Kures for Kids St. Baldrick’s Fellow
Pediatric Brain Tumor Foundation Early Career Development Scholar
The Preston Robert Tisch Brain Tumor Center at Duke
Lab: LSRC B225, 308 Research Dr, Durham, NC 27710 | Tel 919-613-1181
Clinic: 05114A Morris Bldg, Box 3085, 30 Duke Medicine Circle, Durham NC 27710 | Tel 919-668-7342 | Fax 919-668-7345

Comments from Reviewer 1

zjr@duke.edu
Box 3085 DUMC
Durham NC 27710

Tel: 919-668-7336
Fax: 919-668-7345

Referee: The authors have appropriately responded to the previous comments. However, one remaining concern pertains to the experimental validation in HeLa cells. Specifically, while the predictive model has shown a commendable accuracy of 93% when using gold standard data, why is there a substantial discrepancy with the success rate in HeLa cells, which stands at merely 25%?

Authors: Thank you for your thoughtful feedback regarding the experimental validation of our results in HeLa cells. The observed difference in accuracy between METIS and HeLa cells can be attributed to several factors.

Firstly, it's important to note that the 93% accuracy rate was calculated based on all METIS hits, including previously established COMF mutations. These established mutations had a substantial amount of data available in COSMIC, which made their identification by METIS relatively straightforward. In contrast, the 25% identification rate in HeLa cells primarily pertains to the detection of novel and rarer COMF mutations. In 2012, when the METIS1 analysis was conducted, the number of data points for rarer mutations was in the single digits (see Figure 2 in the paper). Consequently, METIS1 was more prone to false positives when dealing with these rarer mutations due to data limitations. We described the steps we took in METIS2, which benefited from a more abundant COSMIC dataset, to address false positives, particularly when dealing with rarer mutations. We now highlight this distinction as follows (page 15, paragraph 3)

“The relatively few cancer mutations included in COSMIC ten years ago limited our ability to implement stricter parameters, which may have resulted in more false positives, as demonstrated by our metabolomic screen only confirming OGDHL A400T as a COMF (Supplemental Figure 7a). In parallel, as METIS’s predictive power increases, we also expect METIS to become more effective at finding COMF mutations.”

Secondly, the discrepancy in accuracy could also be influenced by the sample size used in each scenario. The 25% identification rate in HeLa cells was determined from a sample size of $n=4$, whereas the gold standard panel had a larger sample size of $n=43$. Comparing the accuracy between samples of such differing sizes can be challenging, as smaller sample sizes are inherently more susceptible to variability. That being said, we acknowledge the importance of confirming the existence of COMFs in various cell lineages. For example, while Nadeau et al, 2017 confirmed that CBL Y371H was a COMF mutation in leukemia, it was not validated in HeLa cells. This highlights the need for the validation of COMFs across different cell types, a limitation that we now mention in the discussion as follows (page 21, paragraph 2):

“Confirmation in multiple cancer cell lines would be helpful to catch lineage-specific COMFs, such as CBL Y371H, which was identified in leukemia cells but not the HeLa cells used in this study”

Comments from Reviewer 2

Referee: I thank the authors for considering all comments and critically assessing their implications for the manuscript. All points have been addressed, and the authors have now added adequate comments on several critical aspects of the interpretation of their data.

I support publication of this paper in this revised version.

DukeHealth

Department of Radiation Oncology

THE
PRESTON ROBERT TISCH
BRAIN TUMOR CENTER
at Duke...there is HOPE

Authors: Thank you for your time and consideration in reviewing this manuscript.